# Effects of *Bacillus subtilis* on Growth Performance, Metabolic Profile, and Health Status in Dairy Calves

**DOI:** 10.3390/ani14172489

**Published:** 2024-08-27

**Authors:** Ramūnas Antanaitis, Karina Džermeikaitė, Justina Krištolaitytė, Emilija Armonavičiūtė, Samanta Arlauskaitė, Akvilė Girdauskaitė, Arūnas Rutkauskas, Walter Baumgartner

**Affiliations:** 1Large Animal Clinic, Veterinary Academy, Lithuanian University of Health Sciences, Tilžės Str. 18, LT-47181 Kaunas, Lithuania; karina.dzermeikaite@lsmu.lt (K.D.); justina.kristolaityte@lsmu.lt (J.K.); emilija.armonaviciute@stud.lsmu.lt (E.A.); samanta.arlauskaite@lsmu.lt (S.A.); akvile.dirdauskaite@lsmu.lt (A.G.); arunas.rutkauskas@lsmuni.lt (A.R.); 2Clinical Centre for Ruminant and Camelid Medicine, University of Veterinary Medicine, Veterinaerplatz 1, A-1210 Vienna, Austria; walter.baumgartner@vetmeduni.ac.at

**Keywords:** probiotics, calves, metabolic status, growth performance

## Abstract

**Simple Summary:**

This study evaluated the effects of adding the probiotic *Bacillus (B.) subtilis* to the feed of preweaning neonatal calves. Fifty Holstein calves from the same farm were split into two groups: a control group (CG) fed milk replacer and a treatment group (TG) receiving milk replacer plus 7.5 mL/calf/day of *B. subtilis*. The study began 24 h after birth, with calves’ health monitored daily. Significant differences in body weight were observed between the CG and the TG at 30, 60, and 90 days of age. The TG had a higher average body weight at each time point. At 30 days, the TG also showed lower AST activity and higher GGT activity compared to the CG. Phosphorus levels were higher in the TG, while total protein concentrations were lower at both 30 and 60 days. The inclusion of *B. subtilis* in the calves’ diet led to improved growth and a better metabolic profile, suggesting its potential as a beneficial feed additive in dairy farming.

**Abstract:**

This study focused on assessing whether the inclusion of probiotics (*B. subtilis*) as feed additives during the preweaning stage can enhance the body weight and metabolic condition of neonatal calves. A total of 50 Holstein calves, all born on the same farm, were randomly divided into two homogeneous treatment groups after birth. The calves in the control group (CG) were fed a milk replacer (*n* = 25) (13 females and 12 males) and those in the *B. subtilis*-supplement-treated group (TG), (*n* = 25) (13 females and 12 males) were fed a milk replacer with 7.5 mL/calf/day of *B. subtilis* probiotic (complied with the manufacturer’s guidelines). The probiotic was administered 24 h post-birth, signifying the start of the experimental period. It took one month to collect the animals. Body weight was measured at birth for all animals. A local veterinarian, working on the farm, conducted daily health checks of the calves, recording health parameters and any antibiotic treatments. Blood samples were collected from each calf at birth and 30, 60, and 90 days by puncturing the jugular vein using 10 mL evacuated serum tubes before morning feeding. Significant differences in body weight were observed between the CG and the TG at 30, 60, and 90 days of age. At 30 days, the TG had a 4.11% higher average body weight than the CG (54.38 kg vs. 52.71 kg). At 60 days, the TG’s average weight was 3.75% higher (79.21 kg vs. 76.34 kg), and at 90 days, the TG had a 2.91% higher average weight (112.87 kg vs. 109.67 kg). At 30 days of age, the TG showed significantly lower AST activity, with a 41.12% decrease compared to the CG (51.02 IU/L vs. 72.00 IU/L). Conversely, GGT activity was significantly higher in the TG by 64.68% (40.64 IU/L vs. 14.35 IU/L). Phosphorus concentration at 30 days was also significantly higher in the TG by 9.36% (3.27 mmol/L vs. 2.99 mmol/L). Additionally, the TG had a significantly lower total protein concentration, with a 21.63% decrease at 30 days (46.32 g/L vs. 56.34 g/L) and a 20.28% decrease at 60 days (48.32 g/L vs. 58.12 g/L) compared to the CG. These findings indicate that dairy calves given conventional milk replacer along with a daily dose of 7.5 mL of *B. subtilis* probiotic experienced enhanced growth performance and a more favourable metabolic profile during the first 90 days of their lives.

## 1. Introduction

Many microorganisms, primarily lactic acid bacteria such as *Lactobacillus* spp. and *Bifidobacterium* spp., are recognised as probiotics, along with a smaller number of non-lactic acid bacteria, including *Bacillus (B.) licheniformis* and *B. subtilis* [1].

*Bacillus subtilis* is a thermophilic, spore-producing, Gram-positive, motile bacterium with a rod-shaped morphology [2]. *B. subtilis* is a probiotic strain commonly utilised as a feed supplement for ruminant animals. It can enhance ruminal fermentation and generate a diverse array of enzymes that facilitate intestinal absorption and digestion when administered to cows. In addition to facilitating the digestion of ratio proteins and returning urea to the rumen, *Bacillus subtilis* stimulates protein synthesis, facilitates the absorption of amino acids and peptides generated during decomposition, and augments the protein count that reaches the intestine [3]. *B. subtilis* is capable of synthesising polypeptides that exhibit an antagonistic nature towards intestinal microorganisms, thereby significantly enhancing the digestibility of animal feed. This aerobic bacterium maintains the ecological equilibrium of the intestines by consuming oxygen and contributing to an anaerobic environment. This, in turn, promotes the reproduction of dominant bacteria in the intestines.

Probiotics have been shown to have positive effects on health and the economy [4]. *B. subtilis* is utilised in the feed industry [5] due to its ability to foster favourable changes in intestinal microflora [6], aid in recovery from diarrhoea [7], and enhance average daily growth and feed efficiency [8]. *B. subtilis* is acknowledged as a potent probiotic in the livestock sector. Research by Jeong and Kim [9] highlighted its effectiveness for broilers, including reducing pathogens and boosting beneficial gut bacteria, alongside improving daily growth and feed efficiency. Kritas et al. [10] demonstrated its positive impact on swine, notably enhancing reproductive performance in sows, which includes improved body condition during pregnancy, minimised weight loss during lactation, and increased piglet birth weight. In the case of dairy cows, the use of *B. subtilis* has been linked to increased milk production, as noted by Souza et al. [11]. While the exact mechanism of its beneficial effects is not fully understood, it is theorised that *B. subtilis* may enhance the anaerobic conditions in the gastrointestinal tract by rapidly consuming oxygen during germination, as suggested by Hoa et al. [12]. Feeding *B. subtilis* to calves may lead to enhanced feed efficiency postweaning through the development of beneficial bacteria in the rumen. More extensive research with a larger sample size is required to confirm this potential benefit [13]. *Bacillus subtilis* modifies the rumen microbiome, enhances digestion during weaning, and reduces the intensity of diarrhoea [14]. Bacillus species, unique in their ability to form spores that resist harsh environments and attract scientific interest, have been proposed for oral administration to improve feed efficiency, immune response, and growth performance [1]. Although probiotics derived from Bacillus spp. have been used for more than seven decades, scientific attention has increased dramatically in recent decades [15]. Administering colostrum and milk replacer containing a combination of probiotics and phytobiotics to dairy calves during the preweaning period improved their health, intake, growth, and metabolic status, in contrast to separate additions of phytobiotics (with rosmarinic acid) and multi-strain Lactobacillus probiotics, which showed no significant impact on feed intake, growth, or physiological indices, suggesting a synergistic effect of the combined treatment that requires further investigation in larger field trials [16]. The blend of probiotics and phytobiotics enhanced the metabolic health of dairy calves [16]. An in vitro study indicated the potential of *Bacillus subtilis*, likely through its production of catalase and subtilisin, as suggested by Hosoi et al. [17]. According to Wang et al. [18], future research is needed to confirm the effects of live Bacillus additives in vivo to assess additional live Bacillus additives for enhancing rumen fermentation, and to elucidate the mechanisms by which Bacillus additives affect ruminants. The impact of feeding *B. subtilis* needs to be assessed to more clearly demonstrate the effects of these probiotics [1].

By examining the impact of probiotics on the diversity of intestinal microbiota, we can gain a deeper understanding of the correlation that exists between probiotics, the structure of microbial communities, and overall health [19].

The present study focused on assessing whether the inclusion of probiotics (*Bacillus subtilis*) as feed additives during the first 90 days of age can improve growth performance, metabolic profile, and health status of dairy calves.

## 2. Materials and Methods

### 2.1. Calf Management and Feeding

The research was executed from October 2023 to March 2024 at a dairy farm situated in the western region of Lithuania. The experimental protocols were executed in adherence to the legal requirements of Lithuania. The study received approval under the number PK012858. After birth, a total of 50 (26 males and 24 females) Holstein calves were randomly allocated into two homogeneous treatment groups. All calves came from the same farm. Upon birth, the calves were promptly isolated from their mothers and placed in individual calf hutches in a naturally ventilated barn. It took one month to collect the animals. Every day, the pens were cleaned, and dung was taken away, so they always looked clean and dry. A stomach tube was used to administer 4 L of colostrum to calves within the first hour of life and weighed within 2 h after birth. Culpable colostrum from the mother (≥23 BRIX) was administered to the calves; if the colostrum failed to meet the quality standards, then frozen colostrum was used. Only calves in good health before the study were included in the research. The trial began after birth with the administration of probiotics and lasted until the calf reached 30 days old. Calves were fed individually two times a day. Calves received 3.5 L of milk replacer at each feeding. Replacement milk (Vilomilk-50, Mørke, Denmark) was supplied at 6 A.M. and 6 P.M. The milk substitute was dissolved in hot water (50–60 °C) and administered once the temperature reached around 40 °C. Using a commercial milk replacer (Vilomilk-50, Mørke, Denmark) containing vegetable fat (18.0%), hydrolysed wheat protein concentrate (7.5%), dextrose (2%), vitamins, and trace elements (2.5%), 150 g of milk replacer was combined with one litre of water to make the milk replacer and blended prior to each feeding in accordance with the guidelines provided by the manufacturer. The composition of the milk replacer is presented in Table 1. Throughout the investigation, the experimental feed additive (probiotic) 7.5 mL/calf/day was mixed into the milk substitute just prior to administration. Each calf ingested an equal daily portion of milk replacer. Calves had free access to water. From day 5 to day 30 of the trial, the calves were provided with unlimited access to pelleted starter feed (Table 2).

The weaning procedure was conducted gradually over a span of four days, commencing at 56 days and concluding at 60 days of age.

### 2.2. Creation of Groups

The animals were administered the probiotic twenty-four hours after birth, marking the beginning of the experiment period. The calves in the control group (CG) were fed a milk replacer (*n* = 25) (13 females and 12 males) and the *B. subtilis*-supplement-treated group (TG), (*n* = 25) (13 females and 12 males) was fed a milk replacer with 7.5 mL/calf/day of *B. subtilis* probiotic (concentrated solution of *Bacillus subtilis* probiotics suitable for calf drinking (*Bacillus subtilis* 5 × 107 CFU/mL), UAB Recaidus, Vilniaus g.20–5, Vilnius, Lithuania)). Following delivery, the calves were assigned to either of the two groups in a random manner. Calves in the TG were given probiotics until they were 30 days old, before being weaned. The probiotics were introduced into the morning milk feeding pail.

### 2.3. Measurements

Animals were measured for their weight upon birth and at 15, 30, 60, and 90 days old using a scale (AGRETO Animal Scale, AGRETO Electronics GmbH, Raabs, Austria). The health status of the calves was checked daily by a local veterinarian working on the farm, and health parameters and treatment events (with antibiotics for calves exhibiting clinical symptoms related to bacterial respiratory infections or diarrhoea) were recorded. Calves were monitored daily to assess their health condition. Calves experiencing diarrhoea were administered electrolyte solution, and in cases where it was deemed required, antibiotic treatment was administered. Health and nutritional characteristics, including faecal consistency and body temperature, were assessed. These parameters were analysed based on the research methodology of Salah et al. [20]. We utilised faecal score as a criterion for diarrhoea, following the method outlined by Magalhães et al. [21]. In short, faecal consistency was rated as follows: 1 for hard, 2 for soft or moderate, 3 for runny or mild diarrhoea, and 4 for watery and profuse diarrhoea. Criteria for respiratory disorders were established in accordance with the Wisconsin Health Scoring Grid, as delineated by Vandermeulen et al. [22].

A body temperature over 39.5 °C was classified as a fever. Recorded were gastrointestinal symptoms such as loose stool and respiratory symptoms such as coughing and nasal discharge.

Blood samples were obtained from each calf (*n* = 50) at birth and 30, 60, and 90 days by puncturing the jugular vein using 10 mL evacuated serum tubes to collect blood prior to morning feeding. Blood samples were taken from the jugular vein using an evacuated tube without any anticoagulant (BD Vacutainer^®^, Eysin, Switzerland). The blood samples were transported at a temperature of 4 °C to the Laboratory of Clinical Tests at the Large Animal Clinic of the Veterinary Academy of the Lithuanian University of Health Sciences within one hour of being collected for subsequent examination. Following this, in the laboratory, the blood samples underwent centrifugation for a duration of 10 to 15 min at a speed of 1500× *g*. After feeding colostrum, a portable refractometer (RHC200, Yhequipment, Shenzhen, China) was used to measure the IgG level in the sample serum and look for passive transfer of immunity. This was achieved by placing 1–2 drops of the serum fraction onto a refractometer, following the method reported by Renaud et al. [23]. The activities of gamma-glutamyl transferase (GGT) and aspartate transaminase (AST) and the quantities of iron (Fe), calcium (Ca), magnesium (Mg), phosphorus (P), albumin (ALB), and total protein (TP) were determined in blood serum using a Hitachi 705 analyser (Hitachi, Tokyo, Japan) and DiaSys reagents (Diagnostic Systems GmbH, Berlin, Germany).

### 2.4. Statistical Analysis

The statistical analysis for this study was carried out using IBM SPSS Statistics for Windows, Version 25.0, developed by IBM Corp. in 2017 and based in Armonk, NY, USA. The Shapiro–Wilk test was utilised to ensure the data’s normal distribution. A repeated-measures analysis of variance (ANOVA) test was performed to compare the mean values across the investigated variables. The results were expressed as the mean and standard error of the mean (S.E.M.), with a set significance level of 0.05 (*p* < 0.05) for determining probability.

## 3. Results

### 3.1. Impact of Bacillus Subtilis on Growth of Dairy Calves during the First 90 Days of Life

Significant differences in body weight were observed at 30, 60, and 90 days of age (*p* < 0.05). At 30 days, the control group (CG) averaged 52.71 kg (±10.68), while the test group (TG) averaged 54.38 kg (±12.34), a difference of 1.67 kg, representing a 4.11% increase. At 60 days, the CG averaged 76.34 kg (±2.05) compared to 79.21 kg (±1.86) in the TG, a difference of 2.87 kg, or a 3.75% increase. At 90 days, the CG averaged 109.67 kg (±1.67), whereas the TG averaged 112.87 kg (±1.95), a difference of 3.2 kg, representing a 2.91% increase. There were no significant differences in body weight among the calves at 0 days or 15 days of age (Table 3).

### 3.2. Impact of Bacillus subtilis on the Faecal Score of Dairy Calves During the First 90 Days of Life

During the current study, we did not find any statistically significant differences in faecal scores between the investigated groups. The largest differences were observed at 15 days of age, but they were not significant (Table 4).

### 3.3. Impact of Bacillus subtilis on the Respiratory and Digestive Disorders of Dairy Calves during the First 90 Days of Life

According to our results, we found a higher incidence of digestive diseases in the CG at 15 days of age and an increase in respiratory disorders at 30 days of age (Table 5).

### 3.4. Impact of Bacillus subtilis on Blood Biochemical Parameters of Dairy Calves during the First 90 Days of Life

AST activity. During this study, significantly lower AST activity was observed in the TG in calves at 30 days of age (*p* < 0.001). During this period, the average AST activity in the CG was 72.00 IU/L (±12.68), while in the TG it was 51.02 IU/L (±5.16). The difference in AST activity between the groups was 20.98 IU/L, representing a 41.12% decrease. There were no significant differences in AST activity at 0, 60, or 90 days of age in calves (Table 6).

GGT activity. In this study, we found a statistically significant (*p* < 0.001) increase in gamma-glutamyl transferase activity in the TG compared to the CG at 30 days of age. The GGT activity was 64.68% (or 26.29 IU/L) higher in the TG. The average GGT activity in the TG was 40.64 IU/L (±22.43), while in the CG it was 14.35 IU/L (±3.45). There were no significant differences in GGT activity at 0 days, 60 days, or 90 days of age in calves (Table 7).

Phosphorus. We detected that the average phosphorus concentration in the TG at 30 days of age was significantly (*p* < 0.001) higher by 0.28 mmol/L, corresponding to a 9.36% increase compared to the CG. In the TG, the concentration was 3.27 mmol/L (±0.22), while in the CG it was 2.99 mmol/L (±0.27). There were no significant differences in phosphorus concentration at 0 days, 60 days, or 90 days of age in calves (Table 8).

Total protein. We observed a significant decrease of 21.63% (or 10.02 g/L) in TP concentration in the TG compared to the CG at 30 days of age. The average TP concentration in the TG was 46.32 g/L (±17.43), whereas in the CG it was 56.34 g/L (±9.81). Additionally, we found a significantly lower TP concentration in the TG (average: 48.32 g/L ± 2.29) compared to the CG (average: 58.12 g/L ± 1.55), with a reduction of 9.8 g/L or 20.28% at 60 days of age. There were no significant differences in total protein concentration at 0 days of age in calves (Table 9).

## 4. Discussion

### 4.1. The Influence of Bacillus subtilis on Growth of Dairy Calves during the Initial 90 Days of Life

Statistically significant disparities in body weight were noted at 30, 60, and 90 days of age (*p* < 0.05). According to our results, we found a higher body weight in the TG compared to the CG, with differences of 3.07% at 30 days of age, 3.64% at 60 days of age, and 2.8% at 90 days of age.

Calf growth performance is a crucial metric for assessing the efficacy of feed additives utilised by producers. By functioning as feed additives, probiotics have the capacity to modulate intestinal flora, augment gastrointestinal digestion and absorption, subsequently enhancing animal growth performance and feed conversion rate [24]. Probiotics have the potential to promote intestinal flora balance in animals by stimulating the production of biologically active substances, increasing the abundance of beneficial bacteria, and decreasing the abundance of pathogenic bacteria [25]. Probiotics primarily work by restoring the balance of gut flora and outcompeting harmful bacteria for resources and space, therefore hindering their growth and colonisation. Moreover, probiotics create antimicrobial compounds like organic acids and bacteriocins, regulate the immune system by promoting the production of cytokines and other immune agents, enhance the intestinal epithelial barrier, assist in digestion by producing enzymes that help break down complex molecules, and produce short-chain fatty acids associated with enhanced digestibility and nutrient uptake [26]. Probiotics, widely used as feed additives, play a significant role in forming protective microflora in the gastrointestinal tract of calves. Since the ruminal microorganisms in calves include various bacteria, the hypothesis is that compound probiotics could be beneficial. Therefore, a commercial compound probiotic product was used in this study to assess its impact on Holstein calves [3]. *Bacillus* spp. cultures are utilised in feed additives for enhancing feed digestibility and preventing diseases in industrial cattle husbandry. This practice aims to reduce antibiotic usage, which significantly contributes to animal preservation and productivity. The inclusion of yeasts in feed promotes the growth of anaerobic bacteria, as the respiratory activity of yeasts shields these bacteria in the rumen from damage caused by oxygen [27].

The findings of this study, along with insights from previous research [28,29,30,31,32], indicate that the management and health status of calves play a role in the effectiveness of probiotics on their growth performance. Sun et al. [33] found that the oral administration of *B. subtilis* to calves led to enhanced feed efficiency and an increase in their average daily weight gain. Administering *B. subtilis natto* directly to calves in the preweaning phase improved their growth by enhancing average daily gain and feed efficiency. Additionally, it led to earlier weaning of the calves, all without any negative effects [33]. According to the study of Wu et al., the treated groups had a higher total feed intake compared to the control group, likely due to the enhanced bioavailability of multispecies probiotics that produce organic acids, metabolites, enzymes, and essential nutrients during animal metabolism [34]. In addition, probiotics are thought to indirectly promote the developmental process and fermentation of the reticulorumen. This might lead to higher intake of feed, which may change the microbial population in the rumen [16]. Research on chickens has demonstrated that *Bacillus subtilis* can suppress the growth of pathogenic microorganisms and enhance digestive enzyme activity, leading to a reduction in ammonia production. These effects, in turn, have been shown to improve the growth performance of poultry [33].

Noori et al. [35] discovered that probiotics can enhance the withers height in Holstein calves. This improvement might be attributed to probiotics, which potentially increase the bioavailability of essential minerals like calcium, phosphorus, and magnesium, thereby contributing to the growth of body size. Bacillus species are thought to mitigate pathogen colonisation by activating essential survival pathways and boosting the immune response in epithelial cells, as noted by Williams [36] and Piewngam et al. [37]. However, the impact of probiotics on the growth and health of calves remains uncertain and varies significantly. This variability is often due to differences in the species of probiotics used, the number of viable probiotic bacteria, and the methods of administration, as discussed by Diao et al. [38]. Our study focuses only on body weight data, but it might be improved by including a data analysis of feed intake, rumen fermentation, or bacterial community for a more comprehensive analysis. We recommend including these data in future studies.

### 4.2. The Influence of Bacillus subtilis on the Faecal Score of Dairy Calves in the Initial 90 Days of Life

In the present study, no statistically significant disparities in faecal scores were seen across the groups under investigation. The most notable disparities were noted at 15 days of age, although they were not statistically significant. The faecal score system is a frequently employed method for predicting the occurrence of diarrhoea in young animals [23]. According to Cho and Yoon [39], diarrhoea in preweaning dairy calves is linked to illness, difficulty absorbing nutrients, and reduced productivity in the future. The current investigation revealed that calves experienced several ailments such as diarrhoea, fever, pneumonia, and cough. Diarrhoea is closely associated with inflammation of the gastrointestinal system and has a significant impact on the growth performance of calves [40]. The main problem during the first three weeks of birth is acute diarrhoea, which is then replaced by respiratory disease at about four weeks of age [41]. A study conducted by Medrano-Galarza et al. [42] found that the prevalence of diarrhoea in calves varied between 16% and 27% depending on the season of birth. The average prevalence throughout all seasons was 23%. Furthermore, the prevalence of diarrhoea within a herd was greater than within a pen, with average rates of 23.75% and 17.75%, respectively. The current study found that the incidence of diarrhoea in calves was 12.54%, which was comparatively lower than in previous studies. The health of calves was enhanced by a diet that included compound probiotics, as seen by a reduction in faecal score and the need for medical treatments. Therefore, it is advisable to administer a substantial amount of compound probiotics to enhance the growth and well-being of newly born Holstein calves [3]. Proper feeding management is crucial in preventing viral illnesses and promoting the healthy growth of calves due to their immature immune system [13]. For probiotics to be successfully implemented in calf production, it is critical to identify probiotic bacteria that function as feed additives and to elucidate the mechanisms by which these probiotics influence the intestinal microbiota and immunity of the host to enhance health and performance [43]. The importance of probiotics stems from the significant role a host’s microflora plays in determining susceptibility to diseases, influencing the health of the intestines either positively or negatively [44].

### 4.3. The Influence of Bacillus subtilis on Blood Biochemical Parameters of Dairy Calves during the Initial 90 Days of Life

#### 4.3.1. The Influence of *Bacillus subtilis* on AST Activity of Dairy Calves during the Initial 90 Days of Life

According to our results, the disparity in AST (aspartate aminotransferase) activity across the groups was 20.98 IU/L, indicating a reduction of 41.12% in the TG. Blood AST activities are a crucial metric for assessing liver function, with heightened AST activity signalling liver impairment [45]. AST, an enzyme prevalent in various tissues and organs and particularly active in the liver [46], shows elevated activity in the serum as an indicator of liver or skeletal muscle injury [47]. The levels of hepatic AST rise during pathological events, leading to the release of these enzymes into the bloodstream [48]. Typically, in animals, the generation and neutralisation of free radicals maintain a dynamic equilibrium under normal physiological circumstances, but an overabundance of free radicals can result in oxidative stress [18,49]. According to Wu et al. [34], results with Sprague-Dawley rats involved oral administration of probiotic Bacillus SC06, which decreased activities of alanine transaminase (ALT), AST, alkaline phosphatase (ALP), and lactate dehydrogenase (LDH) and suppressed mitochondrial dysfunction. Studies with lambs and sheep demonstrated reduced activities of AST in the group that was fed *Bacillus subtilis* [50]. In another study by Choonkham et al. [51], it was discovered that cows given additional *B. subtilis* had reduced levels of creatinine and albumin and showed a tendency towards decreased AST and β-hydroxybutyrate concentrations. Additionally, these results clarify the processes through which probiotics mitigate oxidative stress and offer a potentially effective approach to averting liver diseases through the modulation of intestinal microbiota [34].

#### 4.3.2. The Influence of *Bacillus subtilis* on GGT Activity of Dairy Calves during the Initial 90 Days of Life

Our study revealed a significant (*p* < 0.001) rise in GGT activity in the TG compared to the CG at 30 days of age. The GGT activity in the TG was 64.48% greater compared to the reference value of 26.29 IU/L. Mert et al. [52] found that serum GGT activity can be a reliable indicator for assessing passive transfer status in newborn calves. While total protein and urea levels on their own might not provide complete information, they could be useful when evaluated in conjunction with GGT activity. GGT is known to be produced by the cells lining the secretory ducts of the mammary glands. This has led to the understanding that GGT, much like immunoglobulins, is absorbed from the intestines. Previous studies have highlighted that the level of GGT activity in colostrum is significantly higher, by about 400–800 times, than in the serum of adult animals. It has been reported that in calves that consume colostrum, the serum GGT activity is observed to be 60–160 times higher than in adult animals [52].

#### 4.3.3. The Influence of *Bacillus subtilis* on Phosphorus Concentration of Dairy Calves during the Initial 90 Days of Life

Our analysis revealed that the mean phosphorus level in the TG at 30 days old was significantly (*p* < 0.001) elevated by 9.36% compared to the CG. Zhang et al. [1] also found a greater phosphorus concentration in calves fed *Lactobacillus plantarum* and *B. subtilis* compared to the control group. Administering probiotics during the weaning stage may aid in the development of the ruminal bacterial community, as suggested by Krehbiel et al. [53]. This could potentially enhance the utilisation of nutrients [54]. Hosoi et al. [17] observed that in vitro supplementation with *B. subtilis* could enhance the quantity and activity of lactic acid bacteria. However, due to the complex nature of the bacterial community in the rumen, influenced by diet, environmental factors, and the physiological and health status of ruminants, further investigation into rumen fermentation and bacterial community changes is needed. Additionally, assessing the effect of *B. subtilis* alone is crucial for a clearer understanding of the associative impacts of these probiotics [54].

#### 4.3.4. The Influence of *Bacillus subtilis* on Total Protein Concentration of Dairy Calves during the Initial 90 Days of Life

At 30 days of age, we saw a substantial reduction of 17.7% in TP content in the TG when compared to the CG. In addition, we observed a significantly reduced concentration of TP in the TG compared to the CG, with a decrease of 30% at 60 days of age. Most of the probiotics in the milk replacer bypassed the rumen due to the closure of the oesophageal groove, affecting intestinal bacteria. Fuller [55] noted that probiotics are particularly effective in the presence of a microbial population, especially under stress [54]. Administering probiotics during the weaning stage may assist in the growth of the bacterial community within the rumen, according to Krehbiel et al. [53]. This could potentially lead to better nutrient utilisation [54].

Further research, sample collection, and experimentation could be implemented to broaden the scope of this study. It might be beneficial to evaluate blood urea nitrogen (BUN), since it is the primary product of protein metabolism. A higher protein intake results in increased protein breakdown into amino acids, which are then converted into urea and discharged into the bloodstream [56]. Weaned calves supplemented with *Lactobacillus* probiotics showed significantly elevated BUN levels compared to the control group on days 7 and 14, suggesting increased amino acid metabolism and delayed nitrogen excretion [57]. Meanwhile, the relationship between urea nitrogen levels in newborn calf serum and protein intake fluctuates. The fluctuation in blood urea nitrogen levels in calves is influenced by the interplay between their energy levels and protein consumption. Calves may deaminate amino acids to produce carbon skeletons for gluconeogenesis [58]. Exploring the use of plasma urea levels to evaluate protein utilisation in future studies, particularly in calves given probiotics, might be advantageous.

Studying has some limitations. Our study has a relatively small sample size, which might limit the generalisability of the findings. Additionally, the study primarily focuses on short-term outcomes, leaving the long-term effects of *Bacillus subtilis* supplementation on the health and development of calves largely unexplored. 

## 5. Conclusions

The findings indicate that dairy calves who were given conventional milk replacer along with a daily dose of 7.5 mL of *B. subtilis* probiotic experienced enhanced growth performance and a more favourable metabolic profile over the initial 90 days of their lives. The TG exhibited consistently higher body weights compared to the CG at 30, 60, and 90 days of age, with increases of 4.11%, 3.75%, and 2.91%, respectively, indicating enhanced growth. Additionally, the TG demonstrated improved liver health, as evidenced by a 41.12% reduction in AST activity at 30 days, and better protein metabolism, as shown by a 64.68% increase in GGT activity and a 9.36% increase in phosphorus concentration.

Future studies should prioritise increasing the sample size and including additional blood biochemical parameters such as urea, glucose, cholesterol, triglycerides, and others to ensure more reliable and conclusive findings, as well as investigating the long-term impacts of *Bacillus subtilis* supplementation on dairy calves. 

## Figures and Tables

**Table 1 animals-14-02489-t001:** Composition and potency of the milk replacer.

Composition	Amount	Units
Crude protein	22.0	%
Crude fat	18.0	%
Fiber	0.0	%
Ash	6.5	%
Lysine	17.5	g/kg
Methionine	5.0	g/kg
Calcium	7.5	g/kg
Sodium	4.4	g/kg
Phosphorus	6.3	g/kg
Vegetable fat	18.0	%
Hydrolysed wheat protein concentrate	7.5	%
Dextrose	2.0	%
Vitamins and trace elements	2.5	%
**Potency**		
*Bacillus subtilis* concentration	5 × 10^7^ CFU/mL	

**Table 2 animals-14-02489-t002:** Composition of calf starter ingredients.

Nutrient	Amount	Units
Dry matter (DM)	88.82	%
Crude protein (CP)	21.10	% of DM
Soluble protein	27.83	%CP
Ammonium	0.00	% of DM
Acid detergent insoluble protein	0.75	% of DM
Neutral detergent insoluble protein	1.79	% of DM
Acid detergent fibre	10.89	% of DM
Ammonia neutral detergent fibre on an organic matter basis	18.28	% of DM
Physically effective fibre as a percentage of neutral detergent fibre	37.81	% of NDF
Lignin	1.93	% of DM
Non-fibre carbohydrates	47.14	% of DM
Sugar	6.96	% of DM
Starch	23.20	% of DM
Soluble fibre	6.97	% of DM
Ether extract	4.04	% of DM
Ash	9.45	% of DM
Calcium	1.01	% of DM
Phosphorus	0.66	% of DM
Magnesium	0.38	% of DM
Potassium	1.14	% of DM
Sulphur	0.26	% of DM
Natrium	0.50	% of DM
Chloride	0.24	% of DM
Iron	132.59	ppm
Zinc	76.39	ppm
Copper	44.90	ppm
Magnesium	40.88	ppm
Selenium	1.90	ppm
Cobalt	1.90	ppm
Iodide	3.65	ppm
Vitamin A	31.63	IU/kg
Vitamin D	13.53	IU/kg
Vitamin E	288.87	IU/kg

**Table 3 animals-14-02489-t003:** Body weight of dairy calves fed *B. subtilis* supplement (0.5 mL/calf/day) compared to the control group.

BW (kg)	Group	N	Mean	Std. Error	*p*
0 d.	CG	25	34.54	2.05	0.737
	TG	25	35.48	1.86	
15 d.	CG	25	44.21	2.05	0.966
	TG	25	44.35	2.68	
30 d.	CG	25	52.71	2.05	0.048
	TG	25	54.38	2.687	
60 d.	CG	25	76.34	2.05	0.045
	TG	25	79.21	1.86	
90 d.	CG	25	109.67	1.67	0.032
	TG	25	112.87	1.95	

BW—body weigh; 0 d.—body weight at birth; 15 d.—body weight at 15 days of age; 30 d.—body weight at 30 days of age; 60 d.—body weight at 60 days of age; 90 d.—body weight at 90 days of age. CG—control group; TG—*B. subtilis*-supplement-treated group.

**Table 4 animals-14-02489-t004:** Faecal score of dairy calves fed *B. subtilis* supplement (0.5 mL/calf/day) compared to the control group.

FS	Group	N	Mean	Std. Error	*p*
0 d.	CG	25	1.67	0.043	0.87
	TG	25	1.56	0.042	
15 d.	CG	25	2.76	0.034	0.09
	TG	25	2.25	0.095	
30 d.	CG	25	1.54	0.042	0.16
	TG	25	1.67	0.049	
60 d.	CG	25	1.43	0.067	0.65
	TG	25	1.54	0.043	
90 d.	CG	25	1.54	0.076	0.56
	TG	25	1.16	0.056	

FS—faecal score; 0 d.—faecal score at birth; 15 d.—faecal score at 15 days of age; 30 d.—faecal score at 30 days of age; 60 d.—faecal score at 60 days of age; 90 d.—faecal score at 90 days of age. CG—control group; TG—*B. subtilis*-supplement-treated group.

**Table 5 animals-14-02489-t005:** Respiratory and digestive disorders of dairy calves fed *B. subtilis* supplement (0.5 mL/calf/day) compared to the control group.

	Group	N	Number of Respiratory Disorders	Number of Digestive Disorders
0 d.	CG	25	0	2
	TG	25	0	2
15 d.	CG	25	0	11
	TG	25	0	3
30 d.	CG	25	5	2
	TG	25	1	3
60 d.	CG	25	3	1
	TG	25	0	1
90 d.	CG	25	0	1
	TG	25	0	1

0 d.—age at birth; 15 d.—15 days of age; 30 d.—30 days of age; 60 d.—60 days of age; 90 d.—90 days of age. CG—control group; TG—*B. subtilis*-supplement-treated group.

**Table 6 animals-14-02489-t006:** AST activity of dairy calves fed *B. subtilis* supplement (0.5 mL/calf/day) compared to the control group.

AST (IU/L)	Group	N	Mean	Std. Error	*p*
0 d.	CG	25	71.14	6.79	0.35
	TG	25	54.89	2.99	
30 d.	CG	25	72.00	2.53	<0.001
	TG	25	62.02	0.97	
60 d.	CG	25	83.78	2.53	0.543
	TG	25	79.45	0.97	
90 d.	CG	25	66.64	1.53	0.67
	TG	25	65.67	1.32	

AST (IU/L)—aspartate transaminase; 0 d.—AST activity at birth; 30 d.—AST activity at 30 days of age; 60 d.—AST activity at 60 days of age; 90 d.—AST activity at 90 days of age. CG—control group; TG—*B. subtilis*-supplement-treated group.

**Table 7 animals-14-02489-t007:** GGT activity of dairy calves fed *B. subtilis* supplement (0.5 mL/calf/day) compared to the control group.

GGT (IU/L)	Group	N	Mean	Std. Error	*p*
0 d.	CG	25	349.11	81.26	0.168
	TG	25	496.69	66.64	
30 d.	CG	25	14.35	0.72	<0.001
	TG	25	40.64	4.23	
60 d.	CG	25	32.54	0.8	0.154
	TG	25	37.45	1.55	
90 d.	CG	25	25.54	0.8	0.143
	TG	25	28.65	1.55	

GGT (IU/L)—gamma-glutamyl transferase; 0 d.—GGT activity at birth; 30 d.—GGT activity at 30 days of age; 60 d.—GGT activity at 60 days of age; 90 d.—GGT activity at 90 days of age. CG—control group; TG—*B. subtilis*-supplement-treated group.

**Table 8 animals-14-02489-t008:** Phosphorus concentration of dairy calves fed *B. subtilis* supplement (0.5 mL/calf/day) compared to the control group.

P (mmol/L)	Group	N	Mean	Std. Error	*p*
0 d.	CG	25	2.44	0.06	0.745
	TG	25	2.91	0.1	
30 d.	CG	25	2.99	0.05	<0.001
	TG	25	3.27	0.04	
60 d.	CG	25	3.22	0.08	0.54
	TG	25	3.11	0.03	
90 d.	CG	25	3.32	0.04	0.45
	TG	25	3.22	0.06	

P (mmol/L)—phosphorus; 0 d.—phosphorus concentration at birth; 30 d.—phosphorus concentration at 30 days of age; 60 d.—phosphorus concentration at 60 days of age; 90 d.—phosphorus concentration at 90 days of age. CG—control group; TG—*B. subtilis*-supplement-treated group.

**Table 9 animals-14-02489-t009:** Total protein concentration of dairy calves fed *B. subtilis* supplement (0.5 mL/calf/day) compared to the control group.

TP (g/L)	Group	N	Mean	Std. Error	*p*
0 d.	CG	25	47.56	2.01	0.56
	TG	25	49.33	2.29	
30 d.	CG	25	56.34	1.96	0.014
	TG	25	46.32	3.29	
60 d.	CG	25	58.12	1.55	0.045
	TG	25	48.32	2.29	
90 d.	CG	25	60.22	3.55	0.055
	TG	25	55.54	3.29	

TP (g/L)—total protein; 0 d.—total protein concentration at birth; 30 d.—total protein concentration at 30 days of age; 60 d.—total protein concentration at 60 days of age; 90 d.—total protein concentration at 90 days of age. CG—control group; TG—*B. subtilis*-supplement-treated group.

## Data Availability

The original contributions presented in the study are included in the article, further inquiries can be directed to the corresponding author.

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
