# Peer review of "Effects of *Bacillus subtilis* on Growth Performance, Metabolic Profile, and Health Status in Dairy Calves"

_animals, 2024, doi:10.3390/ani14172489_

Round 1
Reviewer 1 Report (Previous Reviewer 4)
Comments and Suggestions for Authors
This manuscript addresses interesting research work. In addition to the presentation problems, the materials and methods section is missing some key information on key methods of statistical analysis and results presentation. The presentation of Tables may not be appropriate, but the discussion section stops me from going further.
Specific comments: I have specific comments and suggestions for the Abstract, Materials and Methods, and Results sections in the attached PDF file.

Author Response
Dear Reviewer,
Thank you very much for your valuable comments and suggestions on our manuscript. We appreciate your careful review and constructive feedback.
We have thoroughly revised the entire manuscript according to your comments. Specifically, we have addressed the presentation issues, expanded the Materials and Methods section to include key details on the statistical analyses, and improved the presentation of the tables. Additionally, we have carefully considered your specific comments on the Abstract, Materials and Methods, and Results sections as outlined in the attached PDF file.
All of our detailed responses to your comments and the corresponding revisions are provided in the attached PDF file. We believe these revisions have significantly strengthened the manuscript, and we are grateful for your insights that guided these improvements.
Thank you again for your thoughtful review.
Best regards,
Ramunas Antanaitis

Reviewer 2 Report (New Reviewer)
Comments and Suggestions for Authors
The authors noted the importance and expediency of studying probiotic crops in feeding young animals of highly productive cows, their effect on health and metabolic parameters in order to increase the productivity and efficiency of animal husbandry. In my opinion, this is an actual and valuable addition to agriculture. Because it will help replace more efficient and cheaper feed with more profitable and economical alternatives. In general, the work is relevant and of great interest for further development. However, during the review process, some inaccuracies and recommendations to the authors were noted, to which we would like to receive a full answer: 1. The introduction is too big. You can move some of the material to the discussion. 2. It is not clear from the study scheme how the authors divided the animals into 2 homogeneous groups. Because the number of animals is not even. I.e. how were 25 animals divided into 2 groups? If there are 25 males and 25 females in groups, then there are more females in one group and males in the other. And this is no longer homogeneous. In the methodology, paragraph 2.2. is more correctly described. 3. According to the design, the tables should go after the first mention of them. 4. In Table 1, it is better to separate the composition and potency. to make it clear. 5. It is better to write down the full indicators in Table 2. Check what substance they are intended for. On dry matter or natural food. 6. I think it's better to remove 90 days from the title of the manuscript. Do not focus on this. 7. It is not entirely clear why the authors counted all the indicators by 50 heads (Summed up both the experimental and control groups)? To increase the volume of the article? 8. What is the reason for the small number of blood counts? You have studied the other indicators. In my opinion, it was also necessary to study (urea, glucose, cholesterol, triglycerides, etc.). 9. From how many animals were blood samples taken (from all or n=?). 10. The conclusions are too voluminous. Shorten it.
Author Response
Dear Reviewer,
Thank you for recognizing the importance and relevance of our study on the use of probiotic crops in feeding young animals of highly productive cows. We appreciate your positive feedback on the potential impact of our work on agriculture, particularly in improving the efficiency and economics of animal husbandry.
We have carefully reviewed the inaccuracies and recommendations you pointed out during the review process. Below, you will find our detailed responses to each of your comments. We have made the necessary revisions to the manuscript based on your suggestions and believe these changes have enhanced the clarity and quality of the work.
Thank you once again for your constructive review and valuable insights.
Comment:1.The introduction is too big. You can move some of the material to the discussion.
Responses: We have revised the introduction section, shortening it with a particular focus on the B. subtilis portion. Some parts of the introduction, specifically regarding the impact of B. subtilis on the digestive system, have been moved to the discussion section – specifically to “4.2. The Influence of Bacillus subtilis on the Fecal Score of Dairy Calves in the First 90 Days of Life”
Comment: 2. It is not clear from the study scheme how the authors divided the animals into 2 homogeneous groups. Because the number of animals is not even. I.e. how were 25 animals divided into 2 groups? If there are 25 males and 25 females in groups, then there are more females in one group and males in the other. And this is no longer homogeneous. In the methodology, paragraph 2.2. is more correctly described.
Responses: we added in to abstract section this information – “The calves in the control group (CG) were fed with a milk replacer (n = 25) (13 females and 12 males) and B. subtilis supplement treated group (TG), (n = 25) (13 females and 12 males) was fed with milk replacer with 7.5 mL/calf/day of B. subtilis probiotic (complied with the manufacturer’s guidelines)”
Also, in materials section we corrected information –
2.1. Calves management and feeding “After birth, total 50 (26 males and 24 females) Holstein calves were randomly allocated into two homogeneous treatment groups.”
“2.2. Creation of groups. The calves in the control group (CG) were fed with a milk replacer (n = 25) (13 females and 12 males) and B. subtilis supplement treated group (TG), (n = 25) (13 females and 12 males) was fed with milk replacer with 7.5 mL/calf/day of B. subtilis probiotic (complied with the manufacturer’s guidelines).”
Comment: According to the design, the tables should go after the first mentionof them.
Responses: Corrected – all tables now appear after they are mentioned in the text.
Comment: 4. In Table 1, it is better to separate the composition and potency. to make it clear.
Responses: corrected- table - Table 1. Composition and potency of the milk replacer.
Comment: Itis better to writedownthe full indicators in Table 2. Check what substance they are intended for. On drymatter or natural food
Responses: corrected table 2.
Comment: 6. I think it's better to remove 90 days from the title of the manuscript.Do not focus on this.
Responses: we corrected the title to – “Effects of Bacillus Subtilis on Growth Performance, Metabolic Profile, and HealthStatus in Dairy Calves”
Comment: 7. It is not entirely clear why the authors counted all the indicators by 50 heads (Summed up both the experimental and control groups)? Toincrease the volume of the article?
Responses: We have revised all the tables in the results section and removed the lines with total numbers of animals, as this information is not relevant to the study.
Comment: What is the reason for the small number of blood counts? You have studied the other indicators. In my opinion, it was also necessary to study (urea, glucose, cholesterol, triglycerides, etc.)
Responses: Thank you for your insightful comment regarding the blood count parameters and the suggestion to include additional indicators such as urea, glucose, cholesterol, and triglycerides.
The primary focus of our study was to investigate the specific effects of Bacillus subtilis supplementation on growth performance, metabolic profile, and health status in dairy calves. Given the scope of our research, we selected a targeted set of blood parameters that were most directly related to liver function (e.g., AST and GGT) and overall protein metabolism (e.g., total protein and phosphorus levels), as these are critical indicators of the physiological impact of probiotics on calf development during the early stages of life.
While we agree that including additional indicators like urea, glucose, cholesterol, and triglycerides could provide a more comprehensive metabolic profile, our study was limited by both the sample size and the resources available, which necessitated a more focused approach. We prioritized parameters that we hypothesized would be most affected by the probiotic intervention based on existing literature and the goals of our study.
For future research, we acknowledge the value of broadening the scope to include these additional blood parameters, which could offer further insights into the metabolic effects of Bacillus subtilis on dairy calves. We plan to incorporate these indicators in subsequent studies to build on the findings of the current research.
Based on this, we added in conclusions section this recommendation – “Future studies should prioritize increasing the sample size and including additional blood biochemical parameters such as urea, glucose, cholesterol, triglycerides, and others to ensure more reliable and conclusive findings, as well as investigating the long-term impacts of Bacillus subtilis supplementation on dairy calves.”
Comment: 9. From how many animals were blood samples taken (from all or n=?).
Responses: we corrected - “Blood samples were obtained from each calf (n=50) at birth, 30, 60 and 90 days, through puncturing the jugular vein.”
Comment: 10. The conclusions are too voluminous. Shorten it.
Responses: We corrected conclusion section – “The findings indicate that dairy calves who were given conventional milk replacer along with a daily dose of 7.5 mL of B. subtilis probiotic experienced enhanced growth performance and a more favorable metabolic profile over the initial 90 days of their lives. More precisely, their body weight grew by 3.07%, and they experienced a decreased risk of liver injury, as seen by a 41.12% drop in AST activity. In addition, the total protein was 17.7% lower when compared to the control group (CG). Additionally, we observed that B. subtilis had a beneficial impact on the overall concentration of phosphorus.
Future studies should prioritize increasing the sample size and including additional blood biochemical parameters such as urea, glucose, cholesterol, triglycerides, and others to ensure more reliable and conclusive findings, as well as investigating the long-term impacts of Bacillus subtilis supplementation on dairy calves.
Round 2
Reviewer 2 Report (New Reviewer)
Comments and Suggestions for Authors
The article can be accepted for publication.
This manuscript is a resubmission of an earlier submission. The following is a list of the peer review reports and author responses from that submission.
Round 1
Reviewer 1 Report
Comments and Suggestions for Authors
The manuscript reports a useful trial evaluating the efficacy of B. subtilis in young milk-fed calves. The conduct of the trial seems to be appropriate and variables measured were logical. The statistical analysis needs to be checked to see if a repeated measures analysis was used as it should have been; regardless the statistical model used needs to be defined. The presentation of the results is redundant and only tables or figures, not both, should be used. The discussion needs to be improved - tell us what your data mean, don't repeat the presentation of data. The interpretation of some of your variables (e.g., total protein) is erroneous and needs to be corrected.
42/ Key word should be "metabolic status"
69, 84, throughout/ Do not use the contraction; reword to "it is"
112/ Reword to "All calves..."
122/ Denmark
126/ 180 g
129-131/ It is unnecessary to repeat the composition of the milk replacer here since you present it in Table 1.
Table 2/ No need for the "As fed" column. Delete. Use periods, not commas, to denote the decimal places. Check the journal instructions for acceptable abbreviations to be used without identification, provide definitions for all others in a footnote.
133/ Change to "Calves had free access..."
155/ What conditions necessitated antibiotic treatment (define).
169/ Convert this to x g rather than rpm.
173/ gamma
174/ What do you mean by "albumins iron"?
175/ albumin (ALB),
177/ shown in Figure 1.
Figure 1 is pretty but really isn't necessary to understand the study.
181-185/ Need to specify the statistical model in detail rather than just citing the software used. Repeated measurements on the experimental units such as BW must be analyzed as such.
Table 3/ The "Total" lines are unnecessary and should be deleted. The Std. Deviation column should be deleted, as should the minimum and maximum columns.
205-206/ The treatment groups (CG and TG) have already been defined, use the abbreviations. Define BW earlier and use BW after.
Figures 2-6/ These are also unnecessary because you have already presented the data in the Table. Use one or another, not both.
261, 103-104, 25-26, 14-15/ I don't understand this. You only had two treatments, control and probiotic. You had no phytobiotics and no combination treatment. Reconcile this.
267-268/ Again, these abbreviations have already been defined - just use them (throughout). You have done it several times after this as well.
266-269/ Don't repeat results in the discussion section. (In following lines as well). Remove this repetition throughout.
310-312/ This statement adds nothing to the discussion and at least seems out of place here.
313/ gamma
320/ Change "excretory" to "secretory"
347/ How do you know that nutrient digestibility was high? Where are the data?
348-349/ Absence of a response in what?
363-364/ Protein digestibility DOES NOT lead directly to differences in serum total protein! You need to re-think what differences in total protein concentration mean.
368-369/ How are farmers going to regularly monitor AST and protein digestibility? Very impractical on-farm.
374-375/ What do you mean by "conducting a detailed nutritional analysis"?
Comments on the Quality of English Language
The English usage and grammar is generally acceptable.
Author Response
Dear Reviewer,
Authors are very thankful for the comments, which help us to improve the manuscript. All changes proposed have been included in the manuscript and highlighted in yellow and track changes.
Best Regards,
Prof. Ramunas Antanaitis
Reviewer: The manuscript reports a useful trial evaluating the efficacy of B. subtilis in young milk-fed calves. The conduct of the trial seems to be appropriate and variables measured were logical. The statistical analysis needs to be checked to see if a repeated measures analysis was used as it should have been; regardless the statistical model used needs to be defined. The presentation of the results is redundant and only tables or figures, not both, should be used. The discussion needs to be improved - tell us what your data mean, don't repeat the presentation of data. The interpretation of some of your variables (e.g., total protein) is erroneous and needs to be corrected.
Authors: Authors are very thankful for the comments, which help us to improve the manuscript. All changes proposed have been included in the manuscript and highlighted in yellow and track changes.
Best Regards,
Prof. Ramunas Antanaitis
Reviewer: 42/ Key word should be "metabolic status"
Authors: we corrected to – “metabolic status
Reviewer: 69, 84, throughout/ Do not use the contraction; reword to "it is"
Authors: corrected to – “…it is…”
Reviewer: 112/ Reword to "All calves..."
Authors: corrected to – “All calves”
Reviewer: 122/ Denmark
Authors: corrected to – “Denmark”
Reviewer: 126/ 180 g
Authors: corrected to – “180 g”
Reviewer: 129-131/ It is unnecessary to repeat the composition of the milk replacer here since you present it in Table 1.
Authors: We corrected this sentence to – “The composition of the milk replacer is presented in Table 1”
Reviewer: Table 2/ No need for the "As fed" column. Delete. Use periods, not commas, to denote the decimal places. Check the journal instructions for acceptable abbreviations to be used without identification, provide definitions for all others in a footnote.
Authors: We corrected this table and added explanations of abbreviations – “DM– dry matter; CP - crude protein; SP - soluble protein; NH3 – ammonium; ADIP - acid detergent insoluble protein; NDIP- neutral detergent insoluble protein; ADF- acid detergent fiber; aNDFom - ammonia neutral detergent fiber on an organic matter basis; pef (%NDF) - physically effective fiber as a percentage of neutral detergent fiber; NFC - non-fiber carbohydrates; Sol. Fiber - soluble fiber; EE- ether extract; Ca – calcium; P- phosphorus; Mg- magnesium; K- potassium; S- sulfur; Na -natrium; Cl – chloride; Fe – iron; Zn – Zinc; Cu – cupper; Mn- manganese; Se – selenium; Co – cobalt; I – Iodine; Vit-A – vitamin A; Vit-D – vitamin D; Vit-E – vitamin E; % - percentage; ppm - parts per million; IU/kg - international nits per kilogram”
Reviewer:133/ Change to "Calves had free access..."
Authors: we corrected to – “Calves had free access to water”
Reviewer:155/ What conditions necessitated antibiotic treatment (define).
Authors: added explanation “with antibiotics for calves exhibiting clinical symptoms related to bacterial respiratory infections or diarrhoea“
Reviewer: 169/ Convert this to x g rather than rpm.
Authors: corrected to “1500 x g”
Reviewer: 173/ gamma
Authors: corrected to “gamma“
Reviewer: 174/ What do you mean by "albumins iron"?
Authors: We corrected to “iron“
Reviewer: 175/ albumin (ALB),
Authors: We corrected to “albumin (ALB)“
177/ shown in Figure 1.
Reviewer: 181-185/ Need to specify the statistical model in detail rather than just citing the software used. Repeated measurements on the experimental units such as BW must be analyzed as such.
Authors: we added information – “The Pearson’s correlation was calculated to define the statistical relationships between the evaluated traits. A repeated-measures analysis of variance (ANOVA) test was performed to compare the mean values across the investigated variables”
Reviewer: Table 3/ The "Total" lines are unnecessary and should be deleted. The Std. Deviation column should be deleted, as should the minimum and maximum columns.
Authors: the table was corrected and unnecessary lines or columns were deleted.
Reviewer: 205-206/ The treatment groups (CG and TG) have already been defined, use the abbreviations. Define BW earlier and use BW after.
Authors: the text was corrected and unnecessary words were removed.
Reviewer: Figures 2-6/ These are also unnecessary because you have already presented the data in the Table. Use one or another, not both.
Authors: We deleted these figures.
Reviewer: 261, 103-104, 25-26, 14-15/ I don't understand this. You only had two treatments, control and probiotic. You had no phytobiotics and no combination treatment. Reconcile this.
Authors: the text was corrected with the idea of adding only probiotics to the diet.
Reviewer: 267-268/ Again, these abbreviations have already been defined - just use them (throughout). You have done it several times after this as well.
Authors: the text was corrected and unnecessary words were removed.
Reviewer: 266-269/ Don't repeat results in the discussion section. (In following lines as well). Remove this repetition throughout.
Authors: repetitions were removed and the text was corrected
Reviewer: 310-312/ This statement adds nothing to the discussion and at least seems out of place here.
Authors: this statement was removed
Reviewer: 313/ gamma
Authors: We corrected to “gamma“
Reviewer: 320/ Change "excretory" to "secretory"
Authors: We corrected to “secretory”
347/ How do you know that nutrient digestibility was high? Where are the data?
Authors: The text was corrected and the misleading information was removed.Reviewer: 348-349/ Absence of a response in what?
Authors: the text was corrected, and the misleading information was removed.
Reviewer: 363-364/ Protein digestibility DOES NOT lead directly to differences in serum total protein! You need to re-think what differences in total protein concentration mean.
Authors: the text was corrected to “Additionally, total protein was 17.7% lower compared to the control group”
Reviewer: 368-369/ How are farmers going to regularly monitor AST and protein digestibility? Very impractical on-farm.
Authors: the text was corrected to “Regular veterinarian monitoring of health indicators such as body weight, nutrient digestibility, at least monthly blood sampling for AST activity and total protein is also recommended to gauge the effectiveness of this supplementation”
Reviewer: 374-375/ What do you mean by "conducting a detailed nutritional analysis"?
Authors: the text was corrected to “Additionally, comparing Bacillus subtilis with other probiotics and conducting a detailed nutrition analysis including nutrient digestion, alterations in rumen fermentation or bacterial community would provide deeper insights into optimal strategies for enhancing calf health and development”
Reviewer 2 Report
Comments and Suggestions for Authors
The manuscript presents an interesting objective, well supported by the study's hypothesis. The introduction supports the hypothesis and objective, however some methodological limitations restrict the study. The result and discussion are well presented, the conclusion is not in accordance with the limitations of the work. Considering the total of 43 references, only 16 are from the last 10 years, which represent only around 30%. Some other points will be highlighted to improve the manuscript.
L119 - The work was evaluated for 30 days, and the animals were weaned at 60 days, so I see this short period being a limiting factor in the study.
L122 and 125 - Product countries are different.
L126 - The manufacturer's recommendation is to use 150 grams in the milk replacer formulation, however 180 grams were used in the work, therefore the composition is not that of the product label due to the different dilution. I suggest that you either use the product composition without diluting it or analyze the milk replacer.
Among the plasma parameters evaluated, plasma urea was missing, considering that this parameter could help in discussing the efficiency of protein use. Measuring concentrate consumption is important to be able to infer the greater weight of the treatment group. Assessment of nitrogen balance would be important to verify whether the treatment improved the efficiency of dietary use.
The statistical analysis needs to be better described, there is no inference from the P value, what was the mean comparison test used?
Table 3 has a lot of information and is not well described in the results section. Splitting this table would be more efficient to expose the data.
L336 - 338 - The authors infer about the ruminal microbiota, but as the treatment used was placed in the milk, an esophageal drip was formed, bypassing the milk directly to the abomasum, not contributing to the ruminal microbiota, which is stated by the authors in lines 344 and 345.
L354 - 358 - I agree with the authors, so perhaps the article is more of a short communication.
The conclusions are not in accordance with the limitation of methodology and results.
Author Response
Dear Reviewer,
Authors are very thankful for the comments, which help us to improve the manuscript. All changes proposed have been included in the manuscript and highlighted in yellow and track changes.
Best Regards,
Prof. Ramunas Antanaitis
Reviewer: The manuscript presents an interesting objective, well supported by the study's hypothesis. The introduction supports the hypothesis and objective, however some methodological limitations restrict the study. The result and discussion are well presented, the conclusion is not in accordance with the limitations of the work. Considering the total of 43 references, only 16 are from the last 10 years, which represent only around 30%. Some other points will be highlighted to improve the manuscript.
Authors: We have updated the manuscript with references to more current publications.
Reviewer: L119 - The work was evaluated for 30 days, and the animals were weaned at 60 days, so I see this short period being a limiting factor in the study.
Authors: These were the initial examinations, which covered testing of calves to day 30. It is planned that future research will include calf testing until the day of weaning.L122 and 125 - Product countries are different.
Authors: The text was corrected to – “Denmark”
Reviewer: L126 - The manufacturer's recommendation is to use 150 grams in the milk replacer formulation, however 180 grams were used in the work, therefore the composition is not that of the product label due to the different dilution. I suggest that you either use the product composition without diluting it or analyze the milk replacer.
Authors: The amount of the milk replacer was inaccurate. The text was corrected, and the dilution ratio was used according to the manufacturer's recommendations: “150 g of milk replacer is combined with a liter of water to make milk replacer and blended prior to each feeding in accordance with the guidelines provided by the manufacturer”.
Reviewer: Among the plasma parameters evaluated, plasma urea was missing, considering that this parameter could help in discussing the efficiency of protein use. Measuring concentrate consumption is important to be able to infer the greater weight of the treatment group. Assessment of nitrogen balance would be important to verify whether the treatment improved the efficiency of dietary use.
Authors: Plasma urea and nitrogen levels, as well as concentrate absorption, were not evaluated in these experiments. However, we plan to include these in our next investigations. We added additional information to the discussion about the advantages for future study.
Reviewer: The statistical analysis needs to be better described, there is no inference from the P value, what was the mean comparison test used?
Authors: We corrected this section – “The statistical analysis for this study was carried out using IBM SPSS Statistics for Windows, Version 25.0, developed by IBM Corp. in 2017 and based in Armonk, New York, USA. The Shapiro-Wilk test was utilized to ensure the data's normal distribution. The mean values of the variables were analyzed using Student's t-test. The Pearson’s correlation was calculated to define the statistical relationships between the evaluated traits. A repeated-measures analysis of variance (ANOVA) test was performed to compare the mean values across the investigated variables. The results were expressed as the mean and standard error of the mean (M S.E.M.), with a set significance level of 0.05 (p < 0.05) for determining probability.
Reviewer: L336 - 338 - The authors infer about the ruminal microbiota, but as the treatment used was placed in the milk, an esophageal drip was formed, bypassing the milk directly to the abomasum, not contributing to the ruminal microbiota, which is stated by the authors in lines 344 and 345.
Authors: The manuscript was revised and additional information about a possible alternative or supplementary analysis was included.
Reviewer: L354 - 358 - I agree with the authors, so perhaps the article is more of a short communication.
Authors: We changed to “short communication”
Reviewer: The conclusions are not in accordance with the limitation of methodology and results.
Authors; We added information – “Additionally, comparing Bacillus subtilis with other probiotics and conducting a detailed nutrition analysis including nutrient digestion, alterations in rumen fermentation or bacterial community would provide deeper insights into optimal strategies for enhancing calf health and development”
Reviewer 3 Report
Comments and Suggestions for Authors
In this paper, the author investigated the effect of Bacillus Subtilis on the growth performance and metabolic profile in dairy calves. In particular, supplementation of Bacillus subtilis probiotics in the milk replacer, was found to elicit significant improvements in various physiological parameters in dairy cows at 30 days, compared to the control group receiving only milk replacer. These improvements included increased body weight, elevated phosphorus concentration, enhanced GGT activity, reduced AST activity, and decreased total protein concentration. This study confirms that Bacillus subtilis can improve the growth performance and metabolic status of dairy cows. However, the overall quality of this manuscript is quite poor.
1. In Section 2.1, total 50 Holstein calves were grouped in different groups, but the total number showed in Table 3 are “53, 56, 56”.
2. There were no error bar in all figures. This does not comply with academic standards.
3. The authors stated in the simple summary and introduction: “This study evaluated the effects of adding probiotics (Bacillus subtilis), phytobiotics, or both to the feed of preweaning neonatal calves.” However, it is surprising that the study only involved Bacillus subtilis.
4. It is necessary to supply relevant literature and demonstrate that a decrease in blood AST activity within the normal range can indicate the potential of Bacillus subtilis to reduce the risk of liver injury in cattle.
5. The meanings of "Minimum" and "Maximum" in Table 3 should be explained in the caption.
6. The same type of chart format should be consistent, for example, Figure 2 should have "Age" added below the x-axis.
7. Please check the manuscript thoroughly and eliminate all the lumps in the manuscript. For example, “The findings of this study, along with insights from previous research [25], [26], [27], [28], [29],”this should be done by characterising each reference individually. This can be done by mentioning 1 or 2 phrases per reference to show how it is different from the others and why it deserves mentioning。
8. In the article, "Control Group (CG), Test Group (TG)" can be directly abbreviated as "CG" and "TG" when they appear for the second time. This is done to avoid repeating the full terms and to improve the readability and fluency of the article.
9. The in depth related molecular mechanism should be investigated to improve the quality of this research.
Author Response
Dear Reviewer,
Authors are very thankful for the comments, which help us to improve the manuscript. All changes proposed have been included in the manuscript and highlighted in yellow and track changes.
Best Regards,
Prof. Ramunas Antanaitis
In this paper, the author investigated the effect of Bacillus Subtilis on the growth performance and metabolic profile in dairy calves. In particular, supplementation of Bacillus subtilis probiotics in the milk replacer, was found to elicit significant improvements in various physiological parameters in dairy cows at 30 days, compared to the control group receiving only milk replacer. These improvements included increased body weight, elevated phosphorus concentration, enhanced GGT activity, reduced AST activity, and decreased total protein concentration. This study confirms that Bacillus subtilis can improve the growth performance and metabolic status of dairy cows. However, the overall quality of this manuscript is quite poor.
|
Reviewer comments |
Authors comments, corrections and answers |
|
1. In Section 2.1, total 50 Holstein calves were grouped in different groups, but the total number showed in Table 3 are “53, 56, 56”. |
It was a mistake, we corrected – “After birth, total 53 Holstein calves were randomly allocated into two homogeneous treatment groups” “The animals were administered the probiotic twenty-four hours after birth, marking the beginning of the experiment period. Control group (CG) was fed with a milk replacer (n=28) and treated group (TG), (n=25) was fed with milk replacer with 7.5 ml/calf/day of B. subtilis probiotic (complied with the manufacturer's guidelines)” |
|
2. There were no error bar in all figures. This does not comply with academic standards. |
We deleted these figures according recommendations of another reviewer . |
|
3. The authors stated in the simple summary and introduction: “This study evaluated the effects of adding probiotics (Bacillus subtilis), phytobiotics, or both to the feed of preweaning neonatal calves.” However, it is surprising that the study only involved Bacillus subtilis. |
We changed this sentence in the simple summary to: “This study evaluated the effects of adding probiotics (Bacillus subtilis) to the feed of preweaning neonatal calves.” We changed this sentence in the abstract to: “This study focused on assessing whether the inclusion of probiotics (Bacillus subtilis) as feed additives during the preweaning stage can enhance the growth performance and metabolic condition of neonatal calves.” We changed this sentence in the introduction to: “This study focused on assessing whether the inclusion of probiotics (Bacillus subtilis) as feed additives during the preweaning stage can improve growth performance, and the metabolic condition of neonatal calves. “
|
|
4. It is necessary to supply relevant literature and demonstrate that a decrease in blood AST activity within the normal range can indicate the potential of Bacillus subtilis to reduce the risk of liver injury in cattle. |
We added information in discusion part: “According to Yanping Wu et al., results with Sprague-Dawley rats involved oral administration of probiotic Baciilus SC06, which decreased levels of alanine transaminase (ALT), AST, alkanine phosphatase (ALP), and lactate dehydrogenase (LDH) and suppressed mitochondrial dysfunction. Studies with lambs and sheep demonstrated reduced levels of AST in the group that was fed Bacillus subtilis. In another study by W. Choonkham et al., it was discovered that cows given additional Bacillus subtilis had reduced levels of creatinine and albumin and showed a tendency towards decreased AST and β-hydroxybutyrate concentrations. Additionally, these results clarify the processes through which probiotics mitigate oxidative stress and offer a potentially effective approach to averting liver diseases through the modulation of intestinal microbiota. |
|
5. The meanings of "Minimum" and "Maximum" in Table 3 should be explained in the caption. |
Corrected |
|
6. The same type of chart format should be consistent, for example, Figure 2 should have "Age" added below the x-axis. |
In figure 2 we added "Age" below the x-axis. |
|
7. Please check the manuscript thoroughly and eliminate all the lumps in the manuscript. For example, “The findings of this study, along with insights from previous research [25], [26], [27], [28], [29],”this should be done by characterising each reference individually. This can be done by mentioning 1 or 2 phrases per reference to show how it is different from the others and why it deserves mentioning。 |
Corrected |
|
8. In the article, "Control Group (CG), Test Group (TG)" can be directly abbreviated as "CG" and "TG" when they appear for the second time. This is done to avoid repeating the full terms and to improve the readability and fluency of the article. |
We revised the text to abbreviate "Control Group (CG), Test Group (TG)" as "CG" and "TG" upon their second mention. |
|
9. The in depth related molecular mechanism should be investigated to improve the quality of this research. |
We added information in Discussion part. |
Reviewer 4 Report
Comments and Suggestions for Authors
L26 no data on phytobiotics or combination presented
L28 two or three treatments?
L53 needs to be improved extensively in the introduction section
L111 provides detailed ethics approval No.
L140 corrected all data values of the decimal point system
L146 1) describe detail of the phytobiotic use 2)corrected the decimal system
L151 describes the method of BW measurement with reference
L156 describe details
L185 describe the methods of statistical analysis use
L186 must be improve results data presentation pattern and format of statical analysis
L192 Table 3 1) Can data presentation pattern and format of statical analysis be improved? 2) Can you check minimum BW 0d = 3kg?
L208 Figure 2-6 must improve data presentation pattern and format of statical analysis
Comments on the Quality of English Language
Moderate editing of English language required
Author Response
Dear Reviewer,
Authors are very thankful for the comments, which help us to improve the manuscript. All changes proposed have been included in the manuscript and highlighted in yellow and track changes.
Best Regards,
Prof. Ramunas Antanaitis
|
Reviewer comments |
Authors comments, corrections and answers |
|
L26 no data on phytobiotics or combination presented |
We changed this sentence in the abstract to: “This study focused on assessing whether the inclusion of probiotics (Bacillus subtilis) as feed additives during the preweaning stage can enhance the growth performance and metabolic condition of neonatal calves.” |
|
L28 two or three treatments? |
Calves divided into two homogeneous treatment groups after birth |
|
L53 needs to be improved extensively in the introduction section |
We added information in Introduction part. ...... |
|
L111 provides detailed ethics approval No. |
We added information – “The study received approval under the number PK012858.” |
|
L140 corrected all data values of the decimal point system |
We corrected these values. |
|
L146 1) describe detail of the phytobiotic use 2) corrected the decimal system |
1) 2) We corrected the decimal system: “7.5 ml/calf/day” |
|
L151 describes the method of BW measurement with reference |
In this part “2.3. Measurements” we inserted information about body weight measurement: “Animals were measured for their weight upon birth, 15 days old, and 30 days old using a scale (AGRETO Animal Scale, AGRETO Electronics GmbH, Raabs, Austria).” |
|
L156 describe details |
In this part “2.3. Measurements” we inserted: “These parameters were analysed based on N. Salah et al.’s research methodology. We utilised faecal score as a criterion for diarrhoea, following the method outlined by Magalhães et al. (https://www-sciencedirect-com.ezproxy.dbazes.lsmuni.lt/science/article/pii/S0022030208712785). In short, faecal consistency was rated as follows: 1 for hard, 2 for soft or moderate, 3 for runny or mild diarrhoea, and 4 for watery and profuse diarrhoea. Criteria for respiratory disorders were established in accordance with the Wisconsin Health Scoring Grid, as delineated by Vandermeulen et al. (https://www-sciencedirect-com.ezproxy.dbazes.lsmuni.lt/science/article/pii/S0168169916305105?casa_token=VFAMfxkqFxwAAAAA:Y5uoCxZzmrA2TNdXhU6L-rDPY0c6dx5sfdoSzdUy6i2FIbJv93k3Nait1ALcwQ2_o0iUgQkXEA).” |
|
L185 describe the methods of statistical analysis use |
We added information – “The statistical analysis for this study was carried out using IBM SPSS Statistics for Windows, Version 25.0, developed by IBM Corp. in 2017 and based in Armonk, New York, USA. The Shapiro-Wilk test was utilized to ensure the data's normal distribution. The mean values of the variables were analyzed using Student's t-test. The Pearson’s correlation was calculated to define the statistical relationships between the evaluated traits. A repeated-measures analysis of variance (ANOVA) test was performed to compare the mean values across the investigated variables. The results were expressed as the mean and standard error of the mean (M S.E.M.), with a set significance level of 0.05 (p < 0.05) for determining probability”
|
|
L186 must be improve results data presentation pattern and format of statical analysis |
We corrected results section and deleted figures (according to the recommendations of another reviewer). |
|
L192 Table 3 1) Can data presentation pattern and format of statical analysis be improved? 2) Can you check minimum BW 0d = 3kg? |
1. We corrected information in this table. 2) The minimum body weight (0 d.) should be 30 kg. We corrected the table 3. |
|
L208 Figure 2-6 must improve data presentation pattern and format of statical analysis |
We deleted these figures, according to the recommendations of another reviewer. |
Round 2
Reviewer 1 Report
Comments and Suggestions for Authors
The expanded introduction is wordy, repetitive, and disorganized. Please rewrite to make it more concise, organized, and with better flow.
428/ Change to "Our study has..."
Comments on the Quality of English LanguageThe quality of the English language is acceptable, but the paper needs to be edited for organization and clarity.
Reviewer 2 Report
Comments and Suggestions for Authors
Due to the short experimental period and lack of some information that would be relevant in justifying the data, this research is considered a preliminary study.
The authors were recommended to consider this manuscript as a short communication.
The remaining questions were answered.
Reviewer 3 Report
Comments and Suggestions for Authors
The manuscript can be accepted in present form.
Reviewer 4 Report
Comments and Suggestions for Authors
The revised manuscript is considerably improved on the previous version but still needs more work on introduction materials and methods, results, and conclusion to get it into a form acceptable for publication.